# Volatilome and Essential Oil of *Ulomoides dermestoides*: A Broad-Spectrum Medical Insect

**DOI:** 10.3390/molecules26206311

**Published:** 2021-10-19

**Authors:** Paulina J. Cázares-Samaniego, Claudia G. Castillo, Miguel A. Ramos-López, Marco M. González-Chávez

**Affiliations:** 1Coordinación para la Innovación de la Ciencia y la Tecnología (CIACYT), Facultad de Medicina, Universidad Autónoma de San Luis Potosí, Sierra Leona #550, Col. Lomas de San Luis, San Luis Potosí C.P. 78210, Mexico; xavet16@gmail.com; 2Facultad de Química, Universidad Autónoma de Querétaro, Cerro de las Campanas s/n, Col. Las Campanas, Santiago de Querétaro, Querétaro C.P. 76010, Mexico; miguel.angel.ramos@uaq.mx; 3Facultad de Ciencias Químicas, Centro de Investigación y Estudios de Posgrado, Universidad Autónoma de San Luis Potosí, Dr. Manuel Nava Martínez #6, Zona Universitaria, San Luis Potosí C.P. 78210, Mexico

**Keywords:** *Ulomoides dermestoides*, VOCs, HS-SPME, PBET, essential oils, GC-MS, insect

## Abstract

*Ulomoides dermestoides* are used as a broad-spectrum medical insect in the alternative treatment of various diseases. Preliminary volatilome studies carried out to date have shown, as the main components, methyl-1,4-benzoquinone, ethyl-1,4-benzoquinone, 1-tridecene, 1-pentadecene, and limonene. This work focused on the production of metabolites and their metabolic variations in *U. dermestoides* under stress conditions to provide additional valuable information to help better understand the broad-spectrum medical uses. To this end, VOCs were characterized by HS-SPME with PEG and CAR/PDMS fibers, and the first reported insect essential oils were obtained. In HS-SMPE, we found 17 terpenes, six quinones, five alkenes, and four aromatic compounds; in the essential oils, 53 terpenes, 54 carboxylic acids and derivatives, three alkynes, 12 alkenes (1-Pentadecene, EOT1: 77.6% and EOT2: 57.9%), 28 alkanes, nine alkyl disulfides, three aromatic compounds, 19 alcohols, three quinones, and 12 aldehydes were identified. Between both study approaches, a total of 171 secondary metabolites were identified with no previous report for *U. dermestoides*. A considerable number of the identified metabolites showed previous studies of the activity of pharmacological interest. Therefore, considering the wide variety of activities reported for these metabolites, this work allows a broader vision of the therapeutic potential of *U. dermestoides* in traditional medicine.

## 1. Introduction

The tenebrionid *Ulomoides dermestoides* (Fairmaire, 1983) (Coleoptera: Tenebrionidae) (synonyms: Alphitobius; Dermestoides; Martianus dermestoides; Palembus dermestoides) is a darkling beetle endemic to the Indomalaya and Papua regions [1]. It is used as a broad-spectrum medical insect in the alternative treatment of various diseases, such as bronchial asthma, dermatitis, rheumatoid arthritis, hemorrhoids, inflammation and pain in the liver and kidneys, Parkinson’s disease, diabetes mellitus, HIV, and different types of cancer [2,3,4,5]. Studies carried out to date have been preliminary, detecting methyl-1,4-benzoquinone (MBQ), ethyl-1,4-benzoquinone (EBQ), 1-pentadecene, and limonene as the significant volatile organic components (VOCs) that are expelled by the insect in its defense secretions [6,7]. To date, just a few investigations have evaluated the pharmacological activity of organic extracts derived from *U. dermestoides*. Among the biological activities evaluated in these investigations are anti-inflammatory [3], cytotoxic [4], antiproliferative [8], antidiabetic [5], antioxidant and antimicrobial activity [9] without attributing the biological activity found to a specific metabolite. The metabolites reported for *U. dermestoides* do not explain the wide spectrum of the medicinal use of the insect; therefore, a more extensive study of the metabolomics of the insect and its variation due to stimuli is required to explain the wide spectrum of entopharmacological use.

Currently, secondary metabolites of insects are obtained using various methodologies, of which we can highlight organic extraction [10,11], solid-phase extraction (SPE) [12,13], and headspace-solid phase microextraction (HS-SPME) [14,15], in conjunction with gas chromatography (GC) coupled to a flame ionization detector (FID) and/or mass spectrometry (MS) for the identification of metabolites. For *U. dermestoides*, only the VOCs have been evaluated by HS-SPME, with the polydimethylsiloxane/divinylbenzene (PDMS/DVB) fiber [6]. However, it has several limitations due to its inability to capture nonpolar and highly polar compounds; therefore, this fiber does not cover a broad spectrum of compounds. For this reason, it remains to be determined whether *U. dermestoides* can produce other chemically diverse secondary metabolites.

Moreover, like other organisms such as plants and bacteria, *U. dermestoides* would be expected to modify its metabolism if the conditions change, such as when the insect has been ingested. In this sense, the production of *U. dermestoides* metabolites and their metabolic variations under stress conditions would provide additional valuable information to help better understand the broad spectrum of its medical uses. In this work, the number of fibers with different polarities used in HS-SPME was increased to determine a higher number of the compounds present in the volatilome of *U. dermestoides* and the metabolic changes that occurred when the insect was subjected to a stimulus that emulated their consumption. The results led to the procurement of the first essential oils from insects, and their characterization allows us to understand that the metabolomics of the insect is more complex than previously reported, thus justifying the wide spectrum of medicinal uses attributed to *U. dermestoides*.

## 2. Results

### 2.1. VOCs Collection with CAR/PDMS Fiber

To determine greater amounts of VOCs of *U. dermestoides*, the volatilome profiles were analyzed by GC-MS under two stimulus conditions over time with CAR/PDMS and PEG fibers.The compounds identified in treatment 1 (T1) in the initial 5 min showed EBQ as the major component (39.21%), followed by limonene (21.82%), and then MBQ, p-benzoquinone (BQ), α-pinene, and 1-pentadecene with 14.29%, 6.11%, 4.35%, and 4.33%, respectively, of the total. However, over time, the relative percentages of these compounds changed—the relative amounts of EBQ and MBQ gradually decreased to as low as 7.44% and 1.94%, respectively, at 18 h, while the relative amounts of limonene, α-pinene, and 1-pentadecene more than doubled during the same period (Table 1, representative chromatograms: Appendix A).

Adding treatment 2 (T2) to the insects had a notable effect on the metabolite profile starting in the first minutes of exposure (Table 1). Here, at 5 min of incubation, 1-pentadecene was the major compound present, comprising 53.74% of the total, followed by limonene (29.22%), and α-pinene, 1-tridecene, EBQ, and carene, with 8.95%, 2.63%, 1.9%, and 1.15% of the total, respectively. Over time, the relative amount of 1-pentadecene decreased to as low as 22.44% of the total at 18 h of incubation. The concentrations of the rest of these VOCs tended to increase over time, with limonene making up 47.7% of the total and α-pinene, 1-tridecene, EBQ, and carene making up, respectively, 11.85%, 7.1%, 6.71%, and 3.9% of the total at 18 h of exposure. An increase in the type of VOCs released was in fact observed at 1 h of incubation with T2. This increase was particularly important for sesquiterpene compounds.

Most importantly, when the two tested stimuli were compared, the beetles that were subjected to T1 immediately released quinone derivatives and limonene, and their metabolism increased the amounts of 1-pentadecene and terpenes when in the presence of the simulated gastric fluid. These differences were also observed at different incubation points, e.g., at 18 h, when higher concentrations of terpenes such as limonene and pinene were observed for insects subjected to T1.

### 2.2. VOCs Collection with PEG Fiber

As shown in Table 2 (representative chromatograms: Appendix A), when the extraction was performed with PEG fiber on beetles under T1, the most prevalent polar compounds identified at the initial time point were quinones. Here, EBQ was the most abundant quinone, making up 54.25% of the total VOCs, followed by MBQ, EHQ, HQ, BQ, and MHQ, which made up, respectively, 17%, 7.45%, 3.54%, 3.08%, and 1.49%; in addition to quinones, 1-pentadecene and limonene were observed in somewhat significant amounts, making up 5.64% and 1%, respectively, of the total. From the initial time point up to 24 h of assessment, the concentrations of the quinones EBQ, EHQ, and MHQ increased to 58.25%, 12.49%, and 2.96% of the total, respectively, while the levels of 1-pentadecene, HQ, BQ, and limonene compounds decreased. Likewise, at 6 h and 18 h, other monoterpenes such as *cis*-verbenol, verbenone, myrtenol, and perillol were observed, as well as two phenol-type compounds, namely m-cresol and 3,4-dimethylphenol.

Moreover, when the extraction was performed with PEG fibers for beetles under T2, 1-pentadecene (67.62%) was initially the main compound observed, followed by EBQ (11.02%), limonene (10.21%), 1-tridecene (1.94%), MBQ (1.16%), and MHQ (0.464%). As the incubation time was increased, these percentages changed considerably; for example, at 24 h, the concentrations of 1-pentadecene and 1-tridecene decreased considerably and those of the quinone-derived compounds increased two- to three-fold from the initial values. Unlike the results in T1, when the sample was treated with T2, neither *cis*-verbenol, verbenone, and myrtenol monoterpenes nor phenol-like metabolites were detected.

In general terms, *U. dermestoides* under the four analysis conditions produced the same types of molecules: quinones, terpenes, and aliphatic alkenes. Moreover, as reported previously, methyl-1,4-benzoquinone, ethyl-1,4-benzoquinone, limonene, 1-tridecene, and 1-pentadecene were present under all four conditions. On the other hand, other secondary metabolites were identified that had not been previously reported for *U. dermestoides*. These results suggested the need to obtain essential oils under both stimulus conditions to obtain a broader vision of the metabolome of *U. dermestoides*.

### 2.3. Obtention and Characterization of Essential Oils

During sample processing by hydrodistillation, characteristic behaviors of each essential oil were observed, namely the striking color of the distillation water for EOT1 and the absence of this in EOT2. On the other hand, oil drops were visible only in EOT2. The yield of the essential oils EOT1 and EOT2 was 0.19% and 0.9%, respectively. For EOT1, 61 compounds were identified, which corresponds to 93.70%; meanwhile, 87 compounds, which represented 92.98% of the total were identified in EOT2.

In both oils, the major component was 1-pentadecene, comprising 77.6% and 57.9% of the total in EOT1 and EOT2, respectively, followed by 1-tridecene (3%), limonene (2.9%), pentacosane (1.8), and tricosane (1.5%), in EOT1, while hentriacontane (6.53%), palmitic acid (6.47%), linoleic acid (2.79%), tricosane (2.79%), pentacosane (2.2%), 1-tridecene (1.75%), oleic acid (1.72%) and limonene (1.43%) were the most abundant compounds in EOT2 (Table 3, Appendix A).

In the essential oils, 11 terpenes, 12 carboxylic acids and their derivatives, 17 alkanes, eight alkenes, nine alkyl disulfides, and three aromatic compounds were identified for the first time in this report. Among the metabolites exclusively found in EOT1, were the terpenes β-thujene and phytan, as well as the alkanes heneicosane and 1-cyclohexyleicosane. On the other hand, squalene, n-hexyl salicylate, ethyl myristate, pentadecanoic acid, linoleic acid, γ-palmitolactone, ethyl stearate, 5-ethyldecane, 6-methylundecano, 3-ethyltetracosane, triacontane, pentacosane, benzothiazole, 6-tert-butyl-3-methylanisole, methyl n-butyl disulfide, ethyl n-butyl disulfide, propyl n-butyl disulfide, butyl n-heptyl disulfide, and pentyl n-heptyl disulfide were found exclusively in EOT2.

Due to the presence of carboxylic acids, alcohols, and aldehydes in essential oils, it was necessary to confirm these results by derivatization. For this analysis, 38 and 77 compounds were identified, which corresponds to an increase of 3.75% and 6.38% in their identification in EOT1 and EOT2, respectively. Therefore, new compounds—25 terpenes, 33 carboxylic acids, and 18 alcohols—were identified with derivatization by silanization (Table 4).

In this analysis, the metabolites found only in EOT1 were 15-isobutyl-(13α-H)-isocopalane, 2-octanoic acid, suberic acid, benzenepropanoic acid, and 2,4-dimethyl-3-pentanol. In the case of EOT2, terpenes as myrtenoic acid, 18-norabieta-8,11,13-triene, 10,18-bisnorabieta-8,11,13-triene, allopregnane, pimaric acid, 7-ethyl-1,4a,7-trimethyl-3,4,4b,5,6,8,10,10a-octahydro-2H-phenanthrene-1-carboxylic acid, isopimaric acid, 8-pimarenic acid, abiet-8-en-18-oic acid, 12α-hydroxy-5α-pregnane, coprostane, cholesterol, germanicol, 3-epimoretenol, campesterol, and avenasterol were found; carboxylic acid and derivatives: butanoic acid, valeric acid, peracetic, 10-undecenoic acid, tridecanoic acid, 2-resorcylic acid, 9-tetradecenoic acid, 9-hexadecenoic acid, *cis*-10-heptadecenoic acid, *cis*-11,14-eicosadienoic acid, *cis*-11-eicosenoic acid, arachidic acid, 1-monopalmitin, docosanoic acid, triacontadienoic acid and dotriacontadienoic acid; and alcohols: 2,2-dimethyl-3-pentanol, furfuryl alcohol, 2,3-butanediol, 4-hydroxybenzenemethanol, 2-pentadecanol, 1-heptadecanol, oleyl alcohol, 1-hexacosanol, 1-octacosanol and 1-dotriacontanol.

In the derivatization for the detection of aldehydes and alkynes, a total of 12 aldehydes and three alkynes were identified. These corresponded to an increase in the total percentage of identified compounds of 0.01% and 0.56% for EOT1 and EOT2, respectively. In summary, the total percentages of identified compounds for EOT1 and EOT2 were 97.46% and 99.92%. The compounds exclusively found in EOT2 were eight of the 12 aldehydes and the three alkynes (Table 5).

In general, the compounds obtained from both oils can be classified into 10 categories: alcohols, aldehydes, alkanes, alkenes, alkynes, alkyl disulfides, aromatic compounds, carboxylic acids, and their derivatives, quinones and terpenes. The amount and type of these metabolites varied depending on the stimulus to which the sample was subjected when the essential oil was obtained. Despite a significant decrease in alkenes and terpenes with respect to the peak area, the variety of these metabolites in EOT2 increased. For the remaining metabolite groups, all compounds increased both in the peak area and in the variety of compounds present (Figure 1).

One of the categories of greatest biological interest is terpenes; therefore, they were analyzed independently. In EOT2, the number of functionalized terpenes increased, and there was a tendency for the peak area to increase with respect to EOT1. Although in terpenes that were not functionalized, the areas of the peaks were smaller, the variety of terpenes present was greater in EOT2 than in EOT1 (Figure 2).

## 3. Discussion

Our results show that the four experimental conditions for the volatilome present three main compound groups: quinones, alkenes, and terpenes. The most abundant compounds are methyl-1,4-benzoquinone, ethyl-1,4-benzoquinone, limonene, 1-pentadecene, and 1-tridecene, in agreement with previous reports [6,7]. However, we found 15 terpenes, four quinones, two alkenes, and four aromatic compounds that had not been previously identified in this organism. The HS-SPME results show the presence of a complex mixture of metabolites of different chemical nature, and the changes over time may be a reflection of the metabolic variety and/or an effect of the equilibrium absorption-desorption process of the compounds in the fiber.

For essential oils, there are five groups of major compounds, these being alkenes (1-pentadecene), carboxylic acids (palmitic, myristic, oleic, and linoleic acids), alkanes (pentacosane and hentriacontane), terpenes (limonene, dehydroabietic acid, β-sitosterol), and alcohols (2-heptanol). 1-Pentadecene is the main component in both essential oils, contrary to the previous report in HS-SPME, where EBQ and MBQ are reported as the main components [6]. This alkene is reported for some coleopters, and it is hypothesized as an epideictic pheromone and defensive secretion [16]. However, other compounds were identified in both analyses, including 50 other terpenes, 37 carboxylic acids, and their derivates, 16 alkanes, nine alkenes, three alkynes, 18 alcohols, 12 aldehydes, nine alkyl disulfides, four quinones, and six aromatic compounds. Note that some terpenes, hydroquinones, and carboxylic acids have been previously reported for other coleopters [17,18,19] but never before, until the current work, for *U. dermestoides*.

Regarding the relative percentage of the identified VOCs, the results obtained in this study suggested a considerably lower rate of release of quinone derivatives by *U. dermestoides* in the presence of the PBET solution than in its absence, and this trend was independent of the type of fiber used for the analysis. However, when analyzing the concentrations of these metabolites in the essential oil, it was observed that their concentration was considerably lower than expected with respect to HS-SPME. This finding indicates that HS-SPME results tend to depend on the balance in the absorption-desorption process, which does not guarantee that the metabolites best captured by the fibers are the most abundant in the sample. In addition, as in other studies [20,21,22], the components identified by hydrodistillation are greater than those obtained by HS-SPME. Therefore, despite the shorter analysis time and the preservation of the sample, HS-SPME remains a limited tool for the characterization of a large number of the compounds present in a complex sample.

However, the fact that quinones are present at low concentrations in essential oils is encouraging, since the importance of these metabolites lies in their potential toxicity. The toxicity of quinones to cells is based on a series of mechanisms that include oxidative stress, redox cycles, arylation, intercalation, induction of cuts in DNA chains, generation of free radicals, and interference with mitochondrial respiration [23,24,25].

In insects, terpenes play essential roles as sex pheromones, trail pheromones, and aggregation and alarm pheromones, as well as in the defense against pathogens [26,27]. It has been postulated that insects can synthesize them de novo, generally as monoterpenes, and they also have the ability to sequester terpenes produced by host plants or endosymbiotic microorganisms [26,27,28]. Monoterpenes are presumably assembled from isopentenyl diphosphate (IDP) and dimethylallyl diphosphate (DMADP) derived from the mevalonate route. In this metabolic pathway, the *trans*- or *cis*- isoprenyl diphosphate synthases (IDSs) catalyze the condensation of IDP with one or two isomers of DMADP [26,27,28,29,30]. *Trans*-IDS enzymes have the particularity of being able to catalyze the syntheses of both precursors and final metabolites, and they can also produce monoterpenes and sesquiterpenes, depending on the cofactor to which they are exposed [29,30,31]. This protein is expressed to a greater extent in insect fat bodies, so this tissue may be the location of the syntheses of terpenes or their precursors [28,32]. Wherever terpenes and/or their precursors are synthesized, they are transported by the hemolymph to reservoir glands, where they are released as part of a defense mechanism [28,29].

Our results suggest that the increase in the size of terpenes produced by *U. dermestoides* could be explained by the release of cofactors that regulate the activity of IDSs via stimulation of simulated gastric juice. The acidic environment produced by this stimulus could improve the bioavailability of metal ions to the beetle and thereby modify the ability of these proteins to regulate activity. Likewise, it has been reported that IDS proteins can remain active over wide ranges of pH (pH 4–8) and temperature (15–45 °C) [29], so they could be active even after the incubation of the beetle with the PBET solution. Although this could tell us how the insect produces terpenes of greater molecular weight, there are no reports in the literature detailing the mechanisms by which an insect modifies the functionalization of the terpenes it produces. This process has been well documented in plants [33]; however, many of the processes and enzymes involved in the metabolomics of insects are still unknown. Likewise, the fact that terpenic acids—which are produced by conifers as well as some species from *Asteraceae*, *Celstraceae*, *Hydrocharitaceae*, and *Lamiaceae*, even some cyanobacterial and fungal species [34,35]—have been detected lays the foundation for rethinking whether the insect not only assimilates these metabolites from food [36,37] but would also be capable of producing them.

As with terpenes, alkanes and alkenes in insects are produced in specialized cells called oenocytes, which are found mainly in the abdomen—associated with epidermal cells or, in some cases, with body fat cells [38,39]. Hydrocarbons are subsequently transported by the hemolymph to both external and internal tissues, including the epicuticle, fat body, ovaries, and reservoir glands [39,40,41]. Cuticle hydrocarbons in insects have two main functions: to protect the insect against desiccation and as signaling molecules in a wide variety of chemical communication systems [42,43].

In our results, a considerable increase in long-chain fatty acids (myristic acid, palmitic acid, stearic acid), saturated aldehydes, and methyl-branched and saturated alcohols were observed in EOT2 compared to EOT1. This could be explained by a modification of the metabolic pathways of cuticle hydrocarbon production by the stress conditions to which the insects were subjected in EOT2 to attempt to protect the insect from the hostile environment to which it was subjected. Therefore, we observe how these precursors (long-chain fatty acids, aldehydes, and alcohols)—as well as the final product of the metabolic route n-alkanes and methyl-branched alkanes—increase. Considering that alkenes have a lower melting point as well as a lower impermeability profile that could affect survival [44], the insect modifies its metabolic routes to enhance the production of alkanes. The absence of the precursors of n-alkenes, unsaturated alcohols, and aldehydes would explain why EOT2 does not increase the number of alkenes identified in the essential oil.

As with the previous metabolites, the increase in the concentration of alkyl disulfides may be a response to the exposure of insects to PBET. However, it is not yet clear how these metabolites are produced or what function they have in the insect.

Many of the metabolites found in *U. dermestoides* have a prior history of clinically important biological activity, among which we can highlight azelaic acid, furfuryl alcohol, benzaldehyde, and phenylacetaldehyde. These compounds possess numerous biological activities of clinical interest, such as anti-inflammatory, antimicrobial, antioxidant, antifungal, and anticancer properties [45,46,47,48]. However, the group of compounds with the greatest diversity of biological activities of interest are terpenes such as limonene, fucosterol, and dehydroabietic acid. These terpenes present antioxidant, anticancer, antiulcer, antihistaminic, antiadipogenic, antiphotodamaging, antimicrobial, antitumor, gastroprotective, hepatoprotective, antiviral, antihyperalgesic, anti-inflammatory, anticholinergic, anti-osteoporotic, antidiabetic, and antihyperlipidemic activities [49,50,51].

## 4. Materials and Methods

### 4.1. Chemicals

The reagents used in this study were sodium citrate, lactic acid, pepsin, N,O-bis(trimethylsilyl)trifluoroacetamide (BSTFA), trimethylsilyl chloride (TMCS), boron trifluoride methanol solution (Sigma-Aldrich, St Louis, MO, USA), DL-malic acid, acetic acid and ethylic ether (JT Baker, Deventer, Holland).

### 4.2. Insects

*Ulomoides dermestoides* were originally obtained from a local provider. The taxonomical identity of *U. dermestoides* was obtained according to the keys published by Kim and Jung (2005) [52], in the Insecticidal Natural Compounds Laboratory of the Faculty of Chemistry, Autonomous University of Querétaro, México. A two-year-old colony was maintained at 27 ± 2 °C and 70 ± 5% relative humidity on a sterile oatmeal substrate and fed with whole bread supplemented with banana peels.

### 4.3. Sample Preparation

Five adult insects were gently placed in a 17 mL glass vial sealed with a Teflon cover with a rubber septum. The samples were incubated and evaluated at five different time points (5 min, 1 h, 6 h, 18 h, and 24 h) in order to monitor changes in the profile of the volatilome of the insect. In addition, the insects were subjected to two treatments, and each condition was tested in duplicate:Treatment 1 (T1): manually shaking the vial for 5 min at room temperature to stimulate the release of defense secretion.Treatment 2 (T2): 1.5 mL of the PBET solution was added to the vial with the insects and incubated at 37 °C with constant agitation (130 rpm). The PBET solution consisted of 0.5 mg/mL sodium citrate, 0.5 mg/mL malic acid, 0.5 µL/mL acetic acid, 0.4 µL/mL lactic acid and ~800 U/mL pepsin at pH 3. The PBET solution simulates the leaching of a solid matrix in the human gastrointestinal tract in order to determine the bioaccessibility of a particular element, such as the total fraction available for adsorption during transit through the small intestine [53]. This digestion simulant solution allowed emulation of the conditions of the insects being ingested and digested by gastric fluid.

### 4.4. VOCs Collection by HS-SPME

The HS-SPME technique was performed using a 75-µm film thickness carboxen/polydimethylsiloxane (CAR/PDMS) and 60-µm film thickness carbowax (PEG) fibers (Supelco, Bellefonte, PA, USA) to detect compounds from nonpolar to polar. To sample the VOCs secreted by *U. dermestoides*, the fibers were placed at a constant distance of 3.4 cm from the insects in treatment 1 and 2.6 cm from the insects in treatment 2. VOCs were absorbed for 15 min, and a desorption time of the fibers of 15 s was used. Fibers were previously conditioned for 5 min at 250 °C at the injection port and reconditioned before each analysis.

### 4.5. Volatilome GS-MS Analysis

GC-MS analysis was performed using a 6890N Network GC System coupled to a 5973 Network mass selective detector (MSD) (Agilent Technologies, Wilmington, DE, USA). The separation was performed using an HP-5MS capillary column (0.25 mm i.d. × 30 m, 0.25 µm film thickness) (J&W, Folsom, CA, USA). The injector was operated in the splitless mode at 250 °C, and the oven temperature was programmed to be 40 °C for 3 min, and then heated at 15 °C/min to 250 °C with a holding time of 5 min at the final temperature. The MSD was operated at 70 eV, the ion source was set at 150 °C, and the transfer line was at 250 °C. VOCs were identified by interpreting their mass spectra fragmentation in the mass range of 50–400 atomic mass units. The software MSD ChemStation (Agilent B.04.02) was used for data recording. The compounds were identified by comparing the obtained mass spectra with those of reference compounds from the National Institute of Standards and Technology (NIST11) and Wiley 9th. The identities of the compounds were confirmed by the Kovats retention index calculated for each peak with reference to the *n*-alkane standards (C7–C18) running under the same conditions.

### 4.6. Obtention of Essential oil of U. dermestoides (EOT1)

An amount of 306 g of adult *U. dermestoides* was hydrodistilled at the boiling temperature of the water. The VOCs were extracted from the stripping water by means of liquid-liquid extraction with ethyl ether. The organic phase was concentrated at 20 °C under reduced pressure until the solvent was eliminated, and the residual water was removed with sodium sulfate.

### 4.7. Obtention of Essential oil of U. dermestoides Post PBET Digestion (EOT2)

An amount of 383 g of adult *U. dermestoides* was digested for 12 h in PBET solution with subsequent inactivation of the solution to pH 7 with sodium bicarbonate. The VOCs were extracted from the stripping water by means of liquid-liquid extraction with ethyl ether. The organic phase was concentrated at 20 °C under reduced pressure until the solvent was eliminated and the residual water was removed with sodium sulfate.

### 4.8. Derivatization for Alcohols and Carboxylic Acid Detection

Essential oils were diluted to 2% in 500 µL heptane and introduced into a 10 mL vial. Then, 100 µL of BSTFA/TMCS solution (9:1 *v*/*v*) was added to the same vial as a silanizing agent. The mixture was reacted at 80 °C under microwave irradiation (200 W microwave power) for 10 min using the Discover System 908,005 (CEM Corporation, NC, USA).

### 4.9. Derivatization for Aldehydes and Alkyne Detection

Essential oils were diluted to 2% in 500 µL heptane and introduced into a 10 mL vial. Then, 100 µL of boron trifluoride 14% in methanol solution was added to the same vial. The mixture was reacted at 80 °C under microwave irradiation (200 W microwave power) for 10 min using the Discover System 908005.

### 4.10. Essential Oil GS-MS Analysis

Samples without derivatization were diluted to 2% in heptanol, using 1 µL of each sample for the analysis, and each sample was analyzed in triplicate. GC-MS analysis was performed using a 7890A Network GC System coupled to a 5975C Network mass selective detector (MSD) and 7683B autosampler (Agilent Technologies, Wilmington, DE, USA). The separation was performed using an HP-5MS capillary column (0.25 mm i.d. × 30 m, 0.25 µm film thickness) (J&W, Folsom, CA, USA). The injector was operated in splitless mode at 300 °C, with a flow of 0.8 mL/min, and the oven temperature was programmed to 40 °C for 3 min, and then heated at 3 °C/min to 300 °C with a holding time of 5 min at the final temperature. The MSD was operated at 70 eV; the ion source was set at 150 °C and the transfer line at 300 °C. VOCs were identified by interpreting their mass spectra fragmentation in the mass range of 15 to 800 atomic mass units. The software MSD ChemStation (Agilent) was used for data recording. The compounds were identified by comparing the obtained mass spectra with those of reference compounds from the National Institute of Standards and Technology (NIST11) and Wiley 9th. The identities of the compounds were confirmed by the Kovats retention index calculated for each peak with reference to the n-alkane standards (C7–C38) running under the same conditions.

### 4.11. Statistical Analysis

The relative percentage of each metabolite was calculated considering the peak area obtained by GC-MS of each metabolite in relation to the total area of peaks analyzed. Data represent the mean of the relative percentage of three repeats ± SD. Metabolites grouped for type for each essential oil were compared with the Mann Whitney U test considering the peak area of each metabolite and a *p* ≤ 0.05. The data in the graphics were expressed as median and range of each group. GraphPad Prism 5 was used to perform the analysis.

## 5. Conclusions

In the volatilome analysis, the use of fibers of different polarities was necessary to expand the detection of metabolites in *U. dermestoides*. Under these analytical conditions, we found 15 terpenes, four quinones, two alkenes, and four aromatic compounds that had not been previously identified in this organism. The composition of essential oils consisted of 10 groups of compounds: alcohols, aldehydes, alkanes, alkenes, alkynes, alkyl disulfides, aromatic compounds, carboxylic acids, and their derivatives, quinones, and terpenes. There were 146 metabolites not previously reported for *U. dermestoides*, in addition to those identified by HS-SPME, of which 76 were found in EOT1 and 132 in EOT2. Between both studies approaches a total of 203 compounds were identified, of which 171 metabolites are reported for the first time in this work for *U. dermestoides*.

In addition, the exposure of *U. dermestoides* to PBET solution in both study approaches showed modifications in the expression of secondary metabolites, principally, an increase in the number of alkanes, alkynes, aromatic compounds, alcohols, alkyl disulfides, carboxylic acids, and terpenoids.

This work reports essential oils obtained from insects for the first time, and also, lays the foundations for the bio-directed study of entopharmacological activity and metabolic pathways of *U. dermestoides* essential oils and their metabolites.

## Figures and Tables

**Figure 1 molecules-26-06311-f001:**
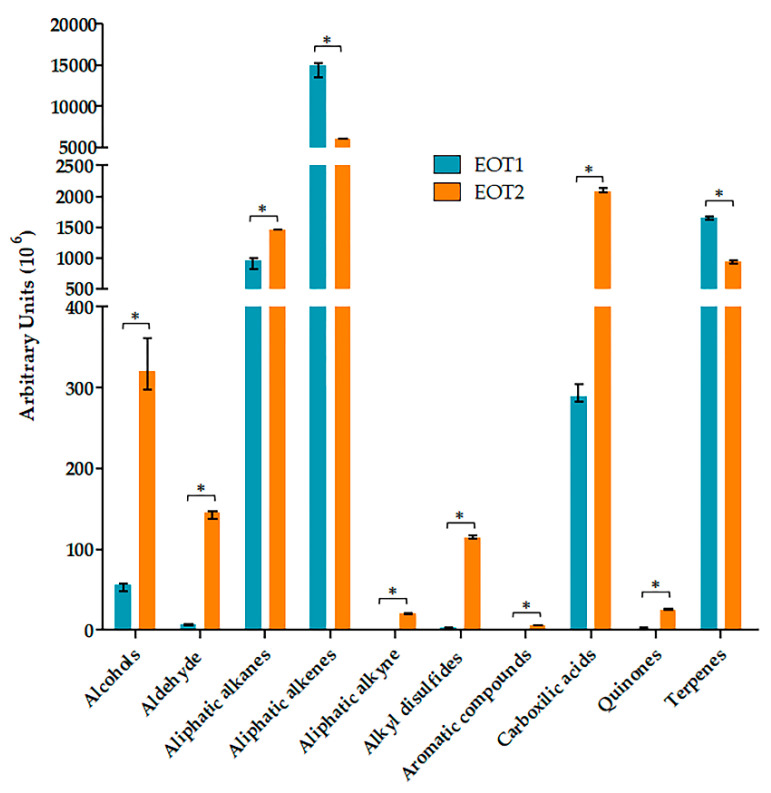
Grouped essential oil compounds. The data are presented as the median of the peak area of each compound (grouped by type) and the range of the data. * Significant difference (*p* ≤ 0.05).

**Figure 2 molecules-26-06311-f002:**
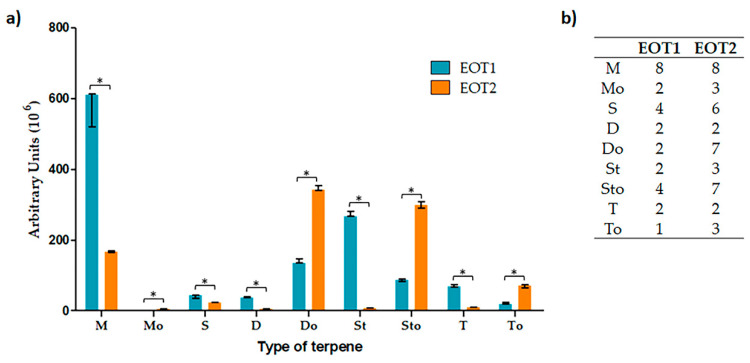
Analysis of the amount and type of terpenes. (**a**) The data are presented as the median of the peak area of each terpene (grouped by type) and the range of the data. M: monoterpene, Mo: monoterpenoids, S: sesquiterpene, D: diterpene, Do: diterpenoids, St: sesterterpenes, Sto: sesterterpenoids, T: triterpene, To: Triterpenoids. * Significant difference (*p* ≤ 0.05). (**b**) Number of terpene compounds in each essential oil.

**Table 1 molecules-26-06311-t001:** VOCs produced by *U. dermestoides* and collected with CAR/PDMS fiber.

No.	Compounds	KI		Relative Abundance (%)									
		Ref	Exp	5 min		1 h		6 h		18 h		24 h	
				T1	T2	T1	T2	T1	T2	T1	T2	T1	T2
* **Quinones** *													
1	*p*-Benzoquinone	912	920	6.11		2.9						0.83	
2	Methyl-1,4-benzoquinone	1015	1016	14.3	0.07	13.6	0.95	3.59	0.68	1.94	1.42	12.1	0.97
3	Ethyl-1,4-benzoquinone	1215	1112	39.2	1.9	33.3	4.89	9.9	2.65	7.44	6.71	34.1	2.56
4	Hydroquinone	1241	1291	0.67		0.11						0.16	
5	2-Methylhydroquinone	1378	1359	0.4		0.37	0.11	0.13				0.65	
6	2-Ethylhydroquinone	1413	1440	1.16		1.08	0.25	0.46		0.18		2.02	
* **Terpenes** *													
1	α-Pinene	922	936	4.35	8.95	7.97	2.35	22.2	6.89	31.6	11.9	6.72	9.09
2	Camphene	958	952	0.13		0.18		0.43		0.75		0.06	
3	Carene	1008	1001	1.27	1.15	1.6	1.24	3.91	2.79	3.58	3.9	1.88	2.94
4	α-Phellandrene	1007	1006	0.26		0.33	0.63	0.62	0.43	0.66	0.59	0.31	0.94
5	*o*-Cymene	1025	1029	0.91	0.67	0.97	0.48	1.8	0.95	1.53	0.92	0.63	1.18
6	Limonene	1020	1033	21.8	29.2	28.4	33	44.4	47.8	36.9	47.7	29.9	42.8
7	γ-Terpinene	1053	1063	0.08		0.05	0.13	0.17	0.12	0.15	0.16	0.06	0.2
8	*p*-Cymenene	1081	1095	0.64		0.55	0.53	0.71	0.33	0.66	0.7	0.32	0.75
9	Di-epi-α-cedrene-(I)	1388	1414	0.15		0.05	0.69	0.08	0.29	0.1		0.1	
10	α-Guaiene	1457	1456				0.11						
11	*cis*-(-)-2,4a,5,6,9a-Hexahydro-3,5,5,9-tetramethyl(1H)benzocycloheptene	1478	1486				0.3						
12	Cuparene	1539	1540	0.09		0.02	1.04	0.03	0.19			0.07	0.37
* **Alkenes** *													
1	1-Tridecene	1287	1295	1.13	2.63	1.17	7.1	1.3	3.24	1.34	2.06	1.02	4.51
2	1,13-Tetradecadiene	1393	1384				0.11						
3	1-Tetradecene	1388	1391				0.64						
4	1,14-Pentadecadiene	1480	1479	0.46		0.19	1.12	0.1	0.45	0.04	0.24	0.42	
5	1-Pentadecene	1486	1494	4.33	53.7	4.63	40.7	9.06	31.7	12	22.4	6.28	31.5
* **Aromatic compounds** *													
1	2,2′-Bifuran	1334	1335	0.25	0.65	0.39		0.44		0.4		0.09	
2	Nonylbenzene	1554	1584				0.18						

RT: Retention time, KI: Kovats index, T1: Agitated only insects, T2: digested with PBET solution insects.

**Table 2 molecules-26-06311-t002:** VOCs produced by *U. dermestoides* and collected with PEG fiber.

No.	Compounds	KI		Relative Abundance (%)									
		Ref	Exp	5 min		1 h		6 h		18 h		24 h	
				T1	T2	T1	T2	T1	T2	T1	T2	T1	T2
* **Quinones** *													
1	*p*-Benzoquinone	912	920	3.08	0.11	0.62						1.29	
2	Methyl-1,4-benzoquinone	1015	1016	17	1.16	15.7	1.46	10.8	1.09	7.42	1.69	17.1	2.61
3	Ethyl-1,4-benzoquinone	1215	1112	54.3	11	54.7	14.7	49	16.9	49.8	22.4	58.5	26.9
4	Hydroquinone	1241	1291	3.54	0.13	0.94		0.21		0.43		2.34	
5	2-Methylhydroquinone	1378	1359	1.49	0.46	2.76	0.61	2.28	1.2	2.55	2.03	2.96	1.47
6	2-Ethylhydroquinone	1413	1440	7.45	4.89	12.1	7.74	15.8	12.3	21.6	22.4	12.5	16.4
* **Terpenes** *													
1	α-Pinene	922	936		0.54		0.5	0.29					0.66
3	Carene	1008	1001		0.19		0.15	0.07					0.17
6	Limonene	1020	1033	1	10.2	1.18	10.5	2.98	10.2	0.64	9.26	0.54	13.6
13	*cis*-Verbenol	1148	1158					0.32		0.19		0.08	
14	*p*-Cymen-8-ol	1172	1196	0.11				0.3		0.16		0.17	
15	Verbenone	1204	1204							0.02			
16	Myrtenol	1213	1212					0.35		0.28			
17	Perillol	1297	1318					0.1					
9	Di-epi-α-cedrene-(I)	1414	1414		0.06								
12	Cuparene	1502	1540		0.66	0.03	0.4	0.13		0.04			
* **Aromatic compounds** *													
3	*m*-Cresol	1053	1088					0.25		2.6			
4	3,4-Dimethylphenol	1167	1180					0.2		2.11			
1	2,2′-Bifuran	1334	1335	0.09									
* **Alkenes** *													
1	1-Tridecene	1287	1295		1.94	0.5	1.37	0.45		0.31		0.47	
4	1,14-Pentadecadiene	1480	1479	3.06	1.04	0.59	0.59	1.14		0.88		0.74	
5	1-Pentadecene	1486	1494	5.64	67.6	10.2	62	14.5	58.3	10.5	42.2	2.16	38.2

**Table 3 molecules-26-06311-t003:** Identified compounds of *U. dermestoides* essential oils.

No.	Compounds	EOUd1		EOUd2		KI Ref
		RA (%)	KI Exp	RA (%)	KI Exp	
Terpenes		4.31%		2.13%		
1	α-Pinene	0.239 ± 0.007	903.6	0.106 ± 0.002	903.6	922
18	β-thujene	0.015 ± 0.000	958.1			968
19	Isolimonene	0.008 ± 0.000	967.6	0.005 ± 0.001	967.6	974
3	2-Carene	0.074 ± 0.002	995.7	0.037 ± 0.001	995.7	996
4	α-Phellandrene			0.002 ± 0.001	1000.2	997
20	α-Terpinene	0.037 ± 0.001	1013	0.084 ± 0.002	1012.9	1008
5	*o*-Cymene	0.029 ± 0.001	1021.3	0.024 ± 0.001	1021.1	1025.4
6	D-Limonene	2.898 ± 0.075	1026.7	1.435 ± 0.016	1025.4	1033
8	*p*-Cymenene	0.006 ± 0.001	1085.1	0.004 ± 0.001	1086.2	1081
21	Terpinen-4-ol	0.002 ± 0.001	1174.9	0.006 ± 0.000	1174.4	1161
14	*p*-Cymen-8-ol	0.002 ± 0.001	1185.1	0.018 ± 0.001	1183.4	1172
9	Di-epi-α-cedrene-(I)	0.213 ± 0.007	1383.1	0.111 ± 0.001	1382.7	1388.2
22	β-Cedrene	0.012 ± 0.001	1418.2	0.005 ± 0.002	1418	1423
23	*cis*-Thujopsene	0.005 ± 0.000	1429.3	0.002 ± 0.001	1428.5	1435
12	Cuparene	0.004 ± 0.002	1518.7	0.075 ± 0.002	1510.9	1502
24	Phytan	0.026 ± 0.002	1807.9			1811
25	Squalene			0.045 ± 0.002	2826.8	2847
26	28-Nor-17β(H)-hopane	0.395 ± 0.022	3044.9	0.038 ± 0.005	3033.9	
27	22R-17alpha(h),21beta(H)-bishomohopane	0.097 ± 0.007	3313.6			
28	γ-Sitosterol	0.248 ± 0.028	3331.9	0.128 ± 0.020	3333.7	3351.3
Alkanes		6.01%		14.74%		
1	4-Propylheptane	0.002 ± 0.000	920.2	0.046 ± 0.002	920.2	945
2	4-Ethyloctane	0.004 ± 0.002	934.4	0.114 ± 0.003	934.4	954
3	4-Methylnonane	0.012 ± 0.001	943.1	0.185 ± 0.003	943.1	963.8
4	5-Methyldecane	0.022 ± 0.013	1056.8	0.181 ± 0.010	1055.2	1057.4
5	Undecane	0.003 ± 0.001	1098.7	0.017 ± 0.001	1098	1100
6	5-Ethyldecane			0.024 ± 0.000	1142.6	1146
7	6-Methylundecane			0.028 ± 0.001	1152.8	1157
8	Dodecane	0.004 ± 0.001	1198.9	0.009 ± 0.001	1198.2	1200
9	Hexadecane	0.020 ± 0.002	1599.4	0.015 ± 0.002	1598.3	1600
10	Heptadecane	0.097 ± 0.009	1701	0.038 ± 0.002	1700.3	1700
11	Octadecane	0.062 ± 0.005	1798.3	0.012 ± 0.001	1797.8	1800
12	Eicosane	0.241 ± 0.011	2001.2	0.166 ± 0.003	2002.4	2000
13	Heneicosane	0.169 ± 0.011	2103.5			2100
14	Docosane	0.161 ± 0.010	2202.2	0.066 ± 0.012	2199.5	2200
15	Tricosane	1.551 ± 0.066	2306.5	2.789 ± 0.017	2302.9	2300
16	Tetracosane	0.545 ± 0.035	2405.2	0.230 ± 0.007	2399.1	2400
17	Pentacosane	1.833 ± 0.084	2511.1	2.270 ± 0.020	2503.5	2500
18	1-Hexadecyloctahydro-1H-indene	0.802 ± 0.068	2553.2			
19	3-Ethyltetracosane			0.015 ± 0.002	2572.8	2567
20	Hexacosane			0.081 ± 0.008	2599	2600
21	Heptacosane	0.324 ± 0.005	2709.6	0.406 ± 0.003	2700	2700
22	1-cyclohexyleicosane	0.123 ± 0.009	2704.2			
23	Octacosane			0.042 ± 0.001	2797.5	2800
24	Nonacosane			0.642±0.006	2899.7	2900
25	Triacontane			0.084±0.013	2996.9	3000
26	Hentriacontane	0.032±0.004	3104.9	6.537±0.015	3107.5	3100
27	Dotriacontane			0.107±0.006	3198.4	3200
28	Tritriacontane			0.638±0.005	3300.3	3300
Alkenes		82.78%		61.51%		
6	Decene	0.003±0.000	985	0.005±0.001	985	987
7	Dodecene	0.006±0.001	1190.8	0.011 ± 0.000	1190.1	1187
1	1-Tridecene	3.020 ± 0.126	1295	1.755 ± 0.006	1293.4	1287
3	1-Tetradecene	0.434 ± 0.019	1392.2	0.231 ± 0.002	1391.6	1385
4	1,14-Pentadecadiene	0.456 ± 0.056	1479.2	0.714 ± 0.007	1477.4	1480
5	1-Pentadecene	77.671 ± 0.906	1517	57.965 ± 0.240	1507.1	1486
8	1-Hexadecene	0.278 ± 0.017	1592.6	0.183 ± 0.006	1591.3	1587
9	(Z,Z)-1,8,11-Heptadecatriene	0.083 ± 0.007	1663.3	0.054 ± 0.002	1662.6	1664.6
10	Heptadecadiene	0.318 ± 0.020	1670.8	0.196 ± 0.002	1670.1	1671
11	Heptadecene	0.513 ± 0.030	1694.4	0.354 ± 0.021	1693.5	1687
12	Pentacosene			0.037 ± 0.001	2473.2	2488
Alkyl disulphides		0.02%		1.16%		
1	Methyl n-butyl disulfide			0.004 ± 0.001	1027.8	1016
2	Ethyl n-butyl disulfide			0.006 ± 0.001	1110.9	1120
3	Propyl n-butyl disulfide			0.004 ± 0.001	1202.2	1207
4	Methyl n-heptyl disulfide	0.007 ± 0.002	1269	0.206 ± 0.001	1268.6	
5	Ethyl n-heptyl disulfide	0.003 ± 0.001	1360.4	0.028 ± 0.003	1359.5	
6	Propyl n-heptyl disulfide	0.002 ± 0.001	1425.6	0.057 ± 0.002	1424.7	
7	Butyl n-heptyl disulfide			0.057 ± 0.001	1524.4	
8	Pentyl n-heptyl disulfide			0.019 ± 0.002	1552.2	
9	Diheptyl disulfide	0.007 ± 0.001	1738.5	0.776 ± 0.003	1738.2	
Aldehydes		0.001%		0.18%		
1	Phenylacetaldehyde	0.001 ± 0.000	1041	0.105 ± 0.004	1040.6	1048
2	Hexadecanal			0.079 ± 0.005	1815.7	1820
Alcohols						
1	1-Heptanol	0.003 ± 0.001	1090	0.007 ± 0.001	1089.4	1092
Quinones		0.00%		0.11%		
3	Ethyl-1,4-benzoquinone			0.083 ± 0.002	1102.6	
6	2-Ethylhydroquinone			0.026 ± 0.003	1437.9	1427
Carboxylic acids and derivatives		0.58%		13.12%		
1	2,4-Dimethyl-5-hexanolide	0.002 ± 0.000	1181.6	0.020 ± 0.001	1180.1	
2	Dodecanoic acid			0.012 ± 0.001	1567.2	1556
3	n-Hexyl salicylate			0.004 ± 0.001	1677.4	1684
4	Myristic acid			0.263 ± 0.019	1767.9	1765
5	Ethyl myristate			0.021 ± 0.002	1794.2	1780
6	Methyl palmitate	0.021 ± 0.001	1929	0.033 ± 0.001	1928.5	1927
7	Pentadecanoic acid			0.065 ± 0.014	1945.8	1942
8	Palmitic acid	0.219 ± 0.031	1967.6	6.475 ± 0.160	1982.2	1964
9	Ethyl palmitate	0.022 ± 0.002	1996.5	0.212 ± 0.002	1996.2	1982
10	Linolenic acid			0.029 ± 0.002	2058.6	2102
11	γ-Palmitolactone			0.086 ± 0.005	2104.4	2106
12	Linoleic acid			2.795 ± 0.142	2148.2	2140
13	Oleic Acid			1.724 ± 0.026	2154	2140
14	Ethyl-9,12-octadecadienoate	0.066 ± 0.008	2165.9	0.345 ± 0.019	2164.2	
15	Ethyl oleate	0.015 ± 0.001	2170.9	0.364 ± 0.069	2170.9	2149
16	Stearic acid	0.077 ± 0.004	2175	0.511 ± 0.013	2173.5	2179
17	Ethyl stearate			0.039 ± 0.008	2196.3	2180
18	Stearyl acetate	0.156 ± 0.017	2213.2	0.119 ± 0.002	2211.3	2211
Aromatic compounds		0.00%		0.04%		
5	Benzothiazole			0.014 ± 0.003	1220.9	1221
6	6-tert-Butyl-3-Methylanisole			0.026 ± 0.001	1235.7	

**Table 4 molecules-26-06311-t004:** Identified compounds of *U. dermestoides* essential oils derivatized by silanization.

No.	Compounds	EOUd1		EOUd2		KI Ref
		RA (%)	KI Exp	RA (%)	KI Exp	
Carboxylic acids and derivatives		0.60%		2.71%		
19	Butanoic acid			0.04 ± 0.005	871.5	891
20	Valeric acid			0.010 ± 0.000	982.4	975
21	Peracetic acid			0.030 ± 0.007	1006	
22	Lactic acid	0.058 ± 0.001	1070.3	0.056 ± 0.001	1072.5	1057
23	Caproic acid	0.009 ± 0.001	1076	0.032 ± 0.002	1078.1	1071
24	2-Ethylhexanoic acid	0.002 ± 0.001	1168	0.011 ± 0.001	1168.7	
25	Heptanoic acid	0.006 ± 0.000	1184.9	0.047 ± 0.002	1185.9	1166
26	Benzoic acid	0.089 ± 0.001	1247	0.028 ± 0.001	1247.4	1232
27	2-Octanoic acid	0.001 ± 0.006	1322.5			1313.2
28	Succinic acid	0.007 ± 0.001	1325.6	0.213 ± 0.027	1325.6	1314
29	Propionylglycine	0.006 ± 0.001	1333.9			1341
30	Nonanoic acid	0.019 ± 0.004	1366.2	0.038 ± 0.002	1366.2	1358
31	Decanoic acid	0.03 ± 0.006	1468.1	0.049 ± 0.003	1467	1455
32	*m*-Hydroxybenzoic acid	0.009 ± 0.001	1528	0.553 ± 0.007	1526.4	1559
33	10-Undecenoic acid			0.006 ± 0.001	1545.2	1542.2
34	Pimelic acid	0.003 ± 0.001	1614.2	0.001 ± 0.000	1614.9	1608
35	Suberic acid	0.012 ± 0.004	1710.3			1689
36	Tridecanoic acid			0.008 ± 0.001	1755.3	1748
37	Azelaic acid	0.061 ± 0.019	1806.9	0.041 ± 0.022	1807.7	1787
38	β-Resorcylic acid			0.005 ± 0.002	1833.9	1822
39	9-Tetradecenoic acid			0.018 ± 0.003	1841.4	
40	Tetradecanoic acid	0.231 ± 0.004	1854.1	1.138 ± 0.025	1856.2	1845
41	Sebacic acid	0.005 ± 0.006	1907	0.002 ± 0.000	1907.5	1920
42	Pentadecanoic acid	0.004 ± 0.001	1925.1	0.015 ± 0.001	1924.8	1942
43	13-methyltetradec-9-enoic acid			0.010 ± 0.001	1946.9	
44	9-Hexadecenoic acid			0.002 ± 0.001	1974.5	1977
45	*cis*-9-Hexadecenoic acid	0.005 ± 0.001	2024.9	0.100 ± 0.003	2023.7	2017
46	*cis*-10-Heptadecenoic acid			0.002 ± 0.001	2127.5	2126.2
47	Margaric acid	0.041 ± 0.006	2152.6	0.085 ± 0.002	2152	2140
48	*cis*-11,14-Eicosadienoic acid			0.036 ± 0.001	2414.8	2413.2
49	*cis*-11-Eicosenoic acid			0.014 ± 0.001	2420.2	2419.7
50	Arachidic acid			0.039 ± 0.002	2447	2437
51	1-Monopalmitin			0.007 ± 0.001	2608.9	2606
52	Docosanoic acid			0.016 ± 0.002	2645	2638
53	Triacontadienoic acid			0.033 ± 0.005	3433.1	
54	Dotriacontadienoic acid			0.025 ± 0.005	3639.9	
Alcohols		0.18%		1.15%		
2	2,2-Dimethyl-3-pentanol			0.025 ± 0.001	993.8	
3	Furfuryl alcohol			0.102 ± 0.017	1003.8	
4	2,4-Dimethyl-3-pentanol	0.011 ± 0.000	1009.8			975.3
5	3-heptanol	0.016 ± 0.001	1018.7	0.095 ± 0.010	1018.7	
6	2-heptanol	0.034 ± 0.014	1025.2	0.394 ± 0.018	1024.8	1008.9
7	2,3-Butanediol			0.282 ± 0.006	1044	1040
1	1-Heptanol	0.004 ± 0.001	1088.4	0.019 ± 0.002	1090	1092
8	3-Ethylphenol	0.001 ± 0.001	1223.1	0.025 ± 0.002	1223.1	1220
9	4-hydroxybenzenemethanol			0.003 ± 0.001	1520.3	1500
10	1-Dodecanol	0.046 ± 0.004	1574.5	0.070 ± 0.003	1574.3	1575
11	1-Tetradecanol	0.039 ± 0.002	1768.5	0.035 ± 0.002	1768.7	1768
12	1-Pentadecanol			0.008 ± 0.000	1868.1	1866
13	2-Pentadecanol			0.001 ± 0.001	1879.4	
14	1-Hexadecanol	0.028 ± 0.003	1966.4	0.022 ± 0.002	1966.6	1965
15	1-Heptadecanol			0.013 ± 0.000	2069.5	2856
16	Oleyl alcohol			0.017 ± 0.003	2136.3	2126
17	1-Hexacosanol			0.007 ± 0.001	2949	2950
18	1-Octacosanol			0.016 ± 0.004	3149	3148
19	1-Dotriacontanol			0.019 ± 0.004	3532.9	3529.9
Quinones		0.01%		0.05%		
4	Hydroquinone	0.008 ± 0.001	1409.7	0.049 ± 0.002	1408.4	1400
Terpenes		2.97%		2.46%		
29	Myrtenoic acid			0.014 ± 0.000	1535.4	
30	18-Norabieta-8,11,13-triene			0.004 ± 0.001	1978.2	
31	10,18-Bisnorabieta-8,11,13-triene			0.014 ± 0.002	2040.9	
32	Allopregnane			0.009 ± 0.003	2204.8	2175
33	Levopimaric acid	0.012 ± 0.001	2262.7	0.015 ± 0.003	2264.6	
34	Pimaric acid			0.020 ± 0.004	2281.7	2287
35	7-Ethyl-1,4a,7-trimethyl-3,4,4b,5,6,8,10,10a-octahydro-2H-phenanthrene-1-carboxylic acid			0.015 ± 0.004	2293.3	
36	15-Isobutyl-(13α-H)-isocopalane	0.110 ± 0.001	2294.2			
37	Isopimaric acid			0.010 ± 0.002	2337.1	2329
38	8-Pimarenic acid			0.102 ± 0.004	2353.8	
39	Abiet-8-en-18-oic acid			0.179 ± 0.003	2371.8	
40	Dehydroabietic acid	0.451 ± 0.011	2394	0.831 ± 0.010	2391.7	2385
41	12α-Hydroxy-5α-pregnane			0.007 ± 0.003	2756	
42	Coprostane			0.010 ± 0.002	2835.6	2822
43	17.alfa.,21β-28,30-Bisnorhopane	0.177 ± 0.010	2873.4	0.005 ± 0.000	2858.8	
44	Gammacerane	0.674 ± 0.031	3135.1	0.019 ± 0.004	3122.8	
45	Cholesterol			0.053 ± 0.002	3151.5	3143
46	Germanicol			0.012 ± 0.001	3208.6	
47	3-Epimoretenol			0.182 ± 0.014	3244.3	
48	Campesterol			0.010 ± 0.001	3259	3220
49	Stigmasterol	0.196 ± 0.014	3296.4	0.090 ± 0.003	3291	3274.3
50	β-Sitosterol	0.991 ± 0.014	3355.2	0.606 ± 0.008	3349.8	3348
51	Fucosterol	0.286 ± 0.004	3370.8	0.197 ± 0.007	3366.3	
52	Aven asterol			0.015 ± 0.001	3421.7	
53	24-Methylenecycloartenol	0.07 ± 0.009	3463.6	0.042 ± 0.004	3459.4	3460
Aromatic compounds		0.001%		0.01%		
7	2,4-Dihydroxyacetophenone	0.001 ± 0.001	1726.1	0.007 ± 0.001	1726.4	1709.3

**Table 5 molecules-26-06311-t005:** Identified compounds of *U. dermestoides* essential oils derivatized by acetal and enol-ether reaction.

	Compounds	EOUd1		EOUd2		KI Ref
		RA (%)	KI Exp	RA (%)	KI Exp	
Aldehydes		0.01%		0.49%		
2	Hexanal	0.001 ± 0.001	971.8	0.013 ± 0.000	968.5	964
3	Heptanal	0.003 ± 0.000	1077.2	0.015 ± 0.001	1077.2	1069
4	Benzaldehyde			0.021 ± 0.001	1107.6	1200
1	Phenylacetaldehyde			0.161 ± 0.001	1217.2	1194
5	Nonanal	0.001±0.001	1278.5	0.041 ± 0.001	1278.9	1267
6	Decanal			0.014 ± 0.002	1377.4	1366
7	Dodecanal			0.013 ± 0.000	1577.4	
8	Tridecanal			0.011 ± 0.003	1676.5	
9	Tetradecanal	0.001 ± 0.001	1774.8	0.051 ± 0.002	1774.8	
10	Pentadecanal			0.012 ± 0.001	1876.8	
11	Hexadecanal			0.068 ± 0.004	1976.9	
12	Octadecanal			0.065 ± 0.010	2177.4	
Alkynes		0.00%		0.07%		
1	Pentadecine			0.009 ± 0.002	1744.4	
2	Hexadecine			0.007 ± 0.000	1849.8	
3	Octadecine			0.054 ± 0.002	2017	

## Data Availability

The representatives chromatograms obtained in this study are available in the Appendix A.

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
