# Peer review of "Volatilome and Essential Oil of Ulomoides dermestoides: A Broad-Spectrum Medical Insect"

_molecules, 2021, doi:10.3390/molecules26206311_

Round 1
Reviewer 1 Report
The paper is well written, it is quite a good and organized study. Introductory story goes well. Material and methods section is written with clarity. Results are presented in a good way and discussion of the results is appropriate. Conclusions are driven by the obtained data. The paper should be published in Molecules journal after minor revisions. Please indicate in conclusion section what is the novelty of the current study and what is the difference of the study with previously published papers on the similar subject,
In my opinion, this manuscript should be published after minor revision without additional review.
Reviewer 2 Report
molecules-1410446
The presented article is very interesting in the field of essential oil chemistry from the insect source. Some minor corrections/modifications are required to improve the quality of the article.
Minor Review
Abstract: please mention the predominant amounts of the identified compounds.
Introduction
Please mention few reported data on the Ulomoides dermestoides regarding chemical and pharmacological works.
Results
Please arrange the compounds according to their chemical classes in all the tables.
Delete line 168-178: no need to mention 16 terpenes. You should only mention which are found in the predominant amount.
No statistical correlation was observed eg. principal component analysis.
Discussion
No comparative discussion was found with the previous data or other species of Volatilome and essential oil components.
Materials and Methods
Please provide the identification number of the species
Conclusions
No future research direction provided

Reviewer 3 Report
This study focused on the metabolite production and metabolic changes of U. dermestoides under stress conditions, providing more valuable information to better understand its broad-spectrum medical use. However, certain parts of the manuscript should be improved:
- Lack of references in materials and methods.
- In Table 1, why does VOCs component content change with time and should be further discussed.
- It is recommended that the total amount of identified ingredients be added to the list of ingredients.
- 1-pentadeceneis the main compound initially observed, and its discussion is not deep enough.
- Line 419 to 422, the author exaggerates the significance of this manuscript in pharmacological action and traditional medicine, which is unacceptable. Because the relationship between bioactivity and compounds needs more specific verification, rather than simple literature citation.
- Figure 2 (b) should be separated as a table
